# Implementation of Individualized Low-Dose Computed Tomography-Guided Hook Wire Localization of Pulmonary Nodules: Feasibility and Safety in the Clinical Setting

**DOI:** 10.3390/diagnostics13203235

**Published:** 2023-10-17

**Authors:** Wei Wei, Shi-Geng Wang, Jing-Yi Zhang, Xiao-Yu Togn, Bei-Bei Li, Xin Fang, Ren-Wang Pu, Yu-Jing Zhou, Yi-Jun Liu

**Affiliations:** Department of Radiology, The First Affiliated Hospital of Dalian Medical University, Dalian 116011, China; weiweidy1988@163.com (W.W.); wangshigeng9855@163.com (S.-G.W.); jingyizhang0330@163.com (J.-Y.Z.); tongxy1226@163.com (X.-Y.T.); 18895663700@163.com (B.-B.L.); fx25632331@163.com (X.F.); purenwang777@163.com (R.-W.P.)

**Keywords:** hook wire, pulmonary nodules, radiation dosage, tomography, X-ray computed

## Abstract

**Background:** CT-guided hook-wire localization is an essential step in the management of small pulmonary nodules. Few studies, however, have focused on reducing radiation exposure during the procedure. **Purpose:** This study aims to explore the feasibility of implementing a low-dose computed tomography (CT)-guided hook wire localization using tailored kVp based on patients’ body size. **Materials and Methods:** A total of 151 patients with small pulmonary nodules were prospectively enrolled for CT-guided hook wire localization using individualized low-dose CT (LDCT) vs. standard-dose CT (SDCT) protocols. Radiation dose, image quality, characteristics of target nodules and procedure-related variables were compared. All variables were analyzed using Chi-Square and Student’s *t*-test. **Results:** The mean CTDIvol was significantly reduced for LDCT (for BMI ≤ 21 kg/m^2^, 0.56 ± 0.00 mGy and for BMI > 21 kg/m^2^, 1.48 ± 0.00 mGy) when compared with SDCT (for BMI ≤ 21 kg/m^2^, 5.24 ± 0.95 mGy and for BMI > 21 kg/m^2^, 6.69 ± 1.47 mGy). Accordingly, the DLP of LDCT was significantly reduced as compared with that of SDCT (for BMI ≤ 21 kg/m^2^, 56.86 ± 4.73 vs. 533.58 ± 122.06 mGy.cm, and for BMI > 21 kg/m^2^, 167.02 ± 38.76 vs. 746.01 ± 230.91 mGy.cm). In comparison with SDCT, the effective dose (ED) of LDCT decreased by an average of 89.42% (for BMI ≤ 21 kg/m^2^) and 77.68% (for BMI > 21 kg/m^2^), respectively. Although the images acquired with the LDCT protocol yielded inferior quality to those acquired with the SDCT protocol, they were clinically acceptable for hook wire localization. **Conclusions:** LDCT-guided localization can provide safety and nodule detection performance comparable to SDCT-guided localization, benefiting radiation dose reduction dramatically, especially for patients with small body mass indexes.

## 1. Introduction

With the widespread application of low-dose computed tomography (LDCT) in recent years, a rapidly increasing number of small and non-determined pulmonary nodules can be possible to detect at an early stage, especially ground glass opacities (GGO), making it vital to manage small pulmonary nodules appropriately [1]. Currently, video-assisted thoracoscopic surgery (VATS) is commonly used for diagnostic and therapeutic purposes for small pulmonary nodules due to its safety and minimal invasive nature [2,3]. Nevertheless, VATS is limited in dealing with most tiny nodules smaller than 10 mm in diameter or more than 5 mm away from the pleural surface [4], which cannot be accurately located by vision, intraoperative finger palpation or endoscopic instrument sliding during the operation, resulting in a prolonged operation time and an increase in the risk of conversion to an open thoracotomy. Specifically, it tends to make the management of pure ground-glass opacity (GGO) with a soft texture more challenging [5]. Therefore, precise location of small pulmonary nodules pre-operation is indispensable, particularly for those located in the lung parenchyma that may be impalpable after deflating the lung. Currently, CT-guided hook wire localization remains the most common preoperative technique for small pulmonary nodules [6,7]. However, the considerable radiation exposure to both patient and interventionalist is an nonnegligible issue, since CT-guided hook-wire localization requires scanning multiple times (planning, guiding and controlling scans) during a single intervention. Thus, the cumulative radiation dose for patients is 40 times greater than that of diagnostic CT examination [8,9].

Previous LDCT studies utilized a uniform scanning protocol for a large population without optimizing the radiation dose based on individual patient body size through the development of personalized scanning protocols [10,11]. Thus, the radiation dose remains an issue that requires further reduction, especially for small-sized patients who are more sensitive to radiation. Applying the scanning parameters for large-sized patients to small ones should be avoided, as it provides no benefit for hook-wire localization, apart from a substantial increase in effective dose. In general, radiation dose is proportional to the square of tube voltage; therefore, a small reduction in tube voltage would result in a significant dose reduction. Theoretically, reducing tube voltage from 120 kVp to 100 kVp or 80 kVp would reduce the radiation dose by 33% and 65%, respectively, assuming all other parameters stay the same [12]. For small-sized patients, as less energy is required for x-ray photon penetration, a relatively low tube voltage is enough to provide sufficient penetration; hence, reducing the radiation dose while preserving image quality [12] is possible. As a result, taking patient’s body size into account, an individualized scanning plan based on body mass index (BMI), lateral diameter or body circumference is useful in routine clinical practice [13,14]. In the majority of people, BMI has been demonstrated to be a reliable predictor of total body fat, and it is one of the most commonly used quantitative indicators to measure body shape [15,16]. Nevertheless, only a few data are available on the radiation dose received by patients with varying BMIs during CT-guided hook wire localization.

This study aims to explore the feasibility of LDCT-guided hook wire localization using tailored kVp based on BMI combined with a fixed mAs of 20.

## 2. Material and Methods

### 2.1. General Information

This study was approved by the Medical Ethics Committee of our hospital, and written informed consent was obtained from all patients before surgery. A prospective analysis was performed on 151 patients from June 2019 to April 2022 who underwent VATS resection after receiving preoperative CT-guided hook wire localization of pulmonary nodules in our hospital. Patient characteristics are shown in Table 1. Exclusion criteria included severe emphysema, rough cough, advanced interstitial pulmonary disease, refractory coagulation dysfunction, cardiac insufficiency, pleurisy and a lesion located close to the mediastinal great vessels.

### 2.2. Scanning Parameters

CT examinations were conducted using a 64-row CT scanner (Somotom Perspective, Siemens™, Erlangen, Germany). The cohort was divided into group A and group B according to the time of enrolment. Group A: 65 patients, standard dose computed tomography (SDCT) scanning from June 2019 to March 2021. Group B: 86 patients, individualized LDCT scans from April 2021 to April 2022. As a part of this study, the routine CT images of patients in group B obtained within one month before surgery were retrospectively collected as group C, with the scanning parameters similar to those of group A, in order to explore the impact of LDCT scanning on the presentation of the same pulmonary nodule. In addition, groups A and B were further divided into four subgroups depending on their BMIs: A1, B1 (BMI ≤ 21 kg/m^2^) and A2, B2 (BMI > 21 kg/m^2^). SDCT scanning was performed using a reference tube voltage of 110 kVp, automated tube current modulation (CARE Dose4D, Siemens Healthcare, Forchheim, Germany) and standard filtered back projection (FBP) reconstruction. In contrast, a LDCT scan was performed using tube voltage of 80 kVp (for BMI ≤ 21 kg/m^2^)/110 kvp (for BMI > 21 kg/m^2^) and 20 mAs as a reference, with the hybrid IR algorithm (SAFIRE, strength level 3) reconstruction. For all groups, the scanning pitch was 1.05, the slice thickness and increment were 5 mm and the reconstruction slice thickness was 1 mm.

### 2.3. Computed Tomography-Guided Hook Wire Localization Procedure

A senior radiologist meticulously reviewed recent chest CT images to determine the optimal puncture route and body position for hook wire placement. Through the use of a radio-opaque grid mesh across the chest wall, CT images were obtained to determine the shape, size and location of the lesion, as well as the interaction with adjacent tissues. Following sterilization and draping of the patient, 2% lidocaine was injected into the chest wall through the needle insertion site. The cannula needle was inserted slowly into the chest wall and lung parenchymal layer, as close to the target nodular lesion as possible, which was typically less than 15 mm away. After determining the appropriate positioning of the introducer needle using the guided CT image, the hook wire was released, and the introducer needle was carefully withdrawn. The CT scan was repeated to confirm the location of the hook wire and to determine whether any complications occurred, including pneumothorax, hemorrhage or subcutaneous emphysema. 

### 2.4. Data Collection

Based on a retrospective analysis, we evaluated the performance of groups B and C in terms of lesion identification. Nodule detection efficiency was compared based on nodule characteristics (CT value and size). On the lung window settings (level, −700; width, 1500), the CT value at the image slice showing largest diameter of the nodule was measured, and the measurement of the largest diameter was also recorded. The region of interest (ROI) for CT value measurement was 3 mm by 3 mm in size, avoiding blood vessels, calcification and necrosis. Characteristics of target nodules (size, location, type, CT values, distance from visceral pleura, shortest distance between the nodule and the puncture needle and distance between the needle tip and the pleura) and procedure-related variables (patient position, number of needle insertions and total procedure time) were recorded. Any signs of pneumothorax, hemorrhage or subcutaneous emphysema would be recorded by the final CT scan or follow-up radiography within two days. The volume CT dose index (CTDIvol) and dose-length product (DLP) were obtained from the dose report. The effective dose (ED) was calculated by multiplying DLP by a conversion factor (0.014 mSv/mGy cm for the chest) [17].

### 2.5. Subjective Imaging Quality Evaluation

Using a 5-point scale, two radiologists with over five years of diagnostic experience independently evaluated each CT image for noise, artifacts, lesion detection, display of anatomical details and contouring. Specific criteria for scoring are as follows: 

5 points for virtually no artifact, good visualization of nodules and anatomical details

4 points for low image noise and the presence of a few artifacts, adequate visualization of nodules and anatomical details

3 points for images with a moderate amount of noise and minor artifacts, a vague visualization of nodules and anatomical details

2 points for a high level of image noise and a moderate level of artifacts, no visualization of nodules and anatomical details 

1 point for the highest level of noise plus the most obvious artifacts, no visualization of nodules and anatomical details. Scores of at least 3 points satisfy the requirements for puncture placement.

### 2.6. Statistical Analysis

Statistical analysis was performed using IBM SPSS Statistics software (Version 26, IBM Corp., Armonk, NY, USA). For continuous variables, the normality of distribution was assessed using the Kolmogorov–Smirnov test. The inter-group comparisons of continuous variables and categorical variables were performed with an independent sample *t*-test and Chi-Square test, respectively. The CT values and nodule sizes in groupsa B and C were analyzed using a paired sample *t*-test. A Mann-Whitney U test was used to analyze the subjective score and radiation dose of group A and B. Kappa statistics was used to evaluate the inter-observer consistency of the subjective scores, and Kappa ≥ 0.75 indicated satisfied consistency. An A value of *p* < 0.05 was considered as statistically significant difference.

## 3. Results

### 3.1. Demographics and Baseline Nodule Characteristics

A total of 151 patients with 184 nodules were enrolled: 52 males and 99 females, with a mean age of 56.3 years (range, 25–75 years). There was no significant demographic difference between the groups (*p* > 0.05). Based on CT scanning, pulmonary nodules could be categorized as pure GGNs (n = 107, 58.2%), part-solid GGNs (n = 67, 36.4%) and solid nodules (n = 10, 5.4%). Other characteristics of patients and nodules are summarized in Table 1.

### 3.2. Localization Procedure-Related Parameters

The intraoperative parameters included the patient’s position, diameter of lesion, lesion-pleura distance, shortest distance between the nodule and the puncture needle, distance between the needle tip and the pleura, number of needle insertions and procedure time and localization-related complications and surgery duration. The values for these parameters obtained for each protocol are summarized in Table 2. There were no significant differences between groups (*p* > 0.05). Figure 1 depicts representative CT-guided lung biopsy images using SDCT versus LDCT.

### 3.3. Evaluation of Nodule Identification Performance

There is no detection efficacy difference between the SDCT and LDCT groups. The CT values of recognizable nodules in the LDCT group ranged from −853.9 HU to 7.2 HU, and nodule sizes ranged from 3.8 mm to 15.5 mm. While the CT values of identifiable nodules in the SDCT group ranged from −753.9 HU to 6.2 HU, and their sizes varied from 3.8 mm to 15.5 mm. No significant difference was observed between LDCT and SDCT in terms of the diameter or CT values of pulmonary nodules (*p* > 0.05) (Table 3). The identification performance of small pulmonary nodules of LDCT and SDCT is shown in Figure 2.

### 3.4. Evaluation of Image Quality

The subjective scores for CT images were comparable between the two observers (*Kappa*: 0.830). The subjective scores of the senior observer were selected for further analysis. Group A received a higher subjective score than group B, and the difference was statistically significant (*p* < 0.001) (Table 4). However, the scores of group B were all greater than 3 points, meeting the requirements for puncture. 

### 3.5. Radiation Dose

As compared to SDCT, the radiation dose for LDCT was significantly reduced (Table 5). The mean CTDIvol was significantly lower for LDCT (for BMI ≤ 21 kg/m^2^, 0.56 ± 0.00 mGy and for BMI > 21 kg/m^2^, 1.48 ± 0.00 mGy) when compared with SDCT (for BMI ≤ 21 kg/m^2^, 5.24 ± 0.95 mGy and for BMI > 21 kg/m^2^, 6.69 ± 1.47 mGy; *p* < 0.0001). Accordingly, the DLP of LDCT was significantly reduced as compared with SDCT (for BMI ≤ 21 kg/m^2^, 56.86 ± 4.73 vs. 533.58 ± 122.06 mGy.cm, and for BMI > 21 kg/m^2^, 167.02 ± 38.76 vs. 746.01 ± 230.91 mGy.cm; *p* < 0.0001). Figure 3 illustrated that the ED value of LDCT group was significantly lower than that of SDCT group, especially in patients with lower BMIs.

## 4. Discussion

In the present study, we explore the feasibility, lesion detection performance and radiation exposure of CT-guided hook wire localization using tailored kVp based on BMI combined with a fixed 20 mAs. Our study demonstrated a significant reduction in the effective dose, from 10.44 ± 3.23 mSv for SDCT intervention to 2.33 ± 0.54 mSv for LDCT intervention (BMI > 21 kg/m^2^) and from 7.47 ± 1.71 mSv for SDCT intervention to 0.79 ± 0.07 mSv for LDCT intervention (BMI ≤ 21 kg/m^2^). This discrepancy represents a reduction of the effective dose by more than 5 to 10 times. 

CT-guided hook wire localization is the most common preoperative technique for small pulmonary nodules. In particular, limiting radiation exposure is extremely important during the procedure of localization, since multiple CT scans are required (planning, guiding and controlling scans). Dose reduction for CT-guided biopsies was normally achieved by decreasing tube voltage or using iterative reconstruction algorithms [18,19,20], and the majority of research relied on comparing DLP or CTDIvol in order to quantify dose reduction. Overall, from any perspective, radiation dose reduction requires further emphasis. To date, a relatively small number of studies was conducted on the use of LDCT in hook wire puncture. Despite some research having achieved low-dose imaging, the effective radiation dose still remains high at 3.17 mSv [21]. The most popular method followed in clinical low-dose scanning scheme was to alter the electrical parameters. Several studies have demonstrated that a lower tube current had no impact on the quality of CT chest image [10,22]. Accordingly, in this study, the tube current of the experimental group was set to an ultra-low level of 20 mAs. As tube voltage has an exponential relationship with radiation dose, lowering tube voltage is a more effective way to achieve the purpose of dose reduction. The tube voltage is also an indicator of radiation penetration ability, so lowering the tube voltage will increase image noise [12]. Thus, when optimizing radiation dose, it is particularly critical to select the appropriate tube voltage in order to balance image noise and ensure image quality. In clinical settings, the selection of tube voltage is strongly influenced by patients’ body size, represented by their BMIs [23,24]. Studies have proved that radiation damage is strongly correlated with patients’ body size [25]. For obese patients, the extra adipose tissue can act as a natural barrier that protects the internal organs. Small and medium-sized patients, however, are at higher risk of radiation damage if they receive the same radiation dose as obese patients. Due to the low radiation dose received by small and medium-sized patients [26], little attention has been paid to their imaging parameters; however, this does not necessarily mean that they have been fully optimized [27]. To further optimize radiation dose, it is necessary to set small and medium-sized patients as a separate category. Therefore, in this study, we conducted a subgroup analysis based on patients’ BMIs. Our results demonstrated that the mean ED of LDCT was significantly reduced compared to SDCT, corresponding to an overall dose reduction of 89.42% (for BMI ≤ 21 kg/m^2^) and 77.68% (for BMI > 21 kg/m^2^), respectively. The ED value for LDCT (BMI > 21 kg/m^2^) was (2.33 ± 0.54) mSv, which was less than 2.5 mSv and classified as low dose imaging, while the ED value of LDCT (BMI ≤ 21 kg/m^2^) was (0.79 ± 0.07) mSv, which was less than 0.8 mSv and categorized as the ultra-low dose computed tomography (uLDCT) imaging [28]. The individualized LDCT protocol for hook wire localization in our study can dramatically reduce the radiation dose of patients, especially for those with low BMIs.

Maintaining image quality and lesion identification performance is crucial in LDCT, as a decreasing radiation dose leads to a rise in image noise. In fact, a previous study explored the feasibility of LDCT lung scanning procedures that decreased radiation exposure while preserving nodule detection performance with an average ED of 1.5 mSv [29]. In the present study, no significant differences in detection efficacy were observed between groups, which means that LDCT (for BMI > 21 kg/m^2^) and uLDCT (for BMI ≤ 21 kg/m^2^) were capable of detecting pulmonary nodules with a similar sensitivity to SDCT. However, multiple comparisons showed the image qualities of LDCT (for BMI > 21 kg/m^2^) and uLDCT (for BMI ≤ 21 kg/m^2^) were significantly poorer than that of SDCT, suggesting that LDCT and uLDCT may be inappropriate for analyzing their imaging characteristics, even though they can detect small pulmonary nodules. Unlike the high standard of image quality for diagnostic purposes, CT-guided hook wire localization only requires complete visualization of the nodule and needle tip [10]. In our investigation, the image quality scores of LDCT and uLDCT were substantially lower than those of SDCT, yet all were greater than three points, meeting the criteria for puncture. 

In the current study, the hook wire was successfully placed with all lung nodules removed through VATS in 151 patients, and none of the procedures needed conversion to thoracotomy. Pneumothorax and hemorrhage are the most common complications associated with hook wire localization [3], along with other factors, such as the number of needle insertions and the duration of the procedure [29]. This study found no statistically significant differences between groups in terms of needle insertions and total puncture time, which is consistent with earlier research [1,9]. The overall frequency of intrapulmonary hemorrhage and pneumothorax did not differ significantly across all groups. All pneumothoraces observed in our study were classified as asymptomatic small pneumothorax [30,31], and none of them required further intervention before surgery. No serious complications such as hemoptysis, massive hemothorax or air embolisms were identified in either group. In light of these findings, it appears that radiation dose reduction does not increase the risk of complications associated with localization.

This research has certain limitations. Firstly, due to the inspection equipment’s limitations, the tube voltage gears were confined to 80 kVp, 110 kVp and 130 kVp. In order to rationally limit the radiation exposure, only 80 kVp and 110 kVp were utilized in this investigation. Secondly, our patient sample size (for BMI ≤ 21 kg/m^2^) was small, thus further clinical trials with larger sample populations are necessary. Thirdly, we were unable to conduct a subgroup analysis focusing on patients who have simultaneous localization for multiple nodules because of the limited number of cases allotted.

## 5. Conclusions

In conclusion, LDCT-guided localization can provide comparable safety and nodule detection performance compared to SDCT-guided localization, while dramatically reducing radiation exposure, especially for patients with small body mass indexes.

## Figures and Tables

**Figure 1 diagnostics-13-03235-f001:**
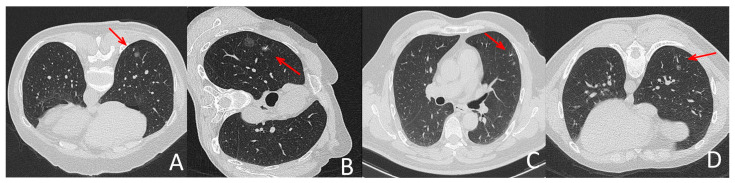
Representative localization images with LDCT and SDCT protocols. (**A**,**C**) were images during SDCT-guided localization, (**B**,**D**) were pictures during LDCT-guided localization. All groups of pictures can clearly show the puncture needle and the lesion. (**B**,**D**) showed that the quality of CT pictures in LDCT group meet the demand of localization for lung nodules in different lobes and patients in different position. The arrow indicates the location of the needle tip.

**Figure 2 diagnostics-13-03235-f002:**
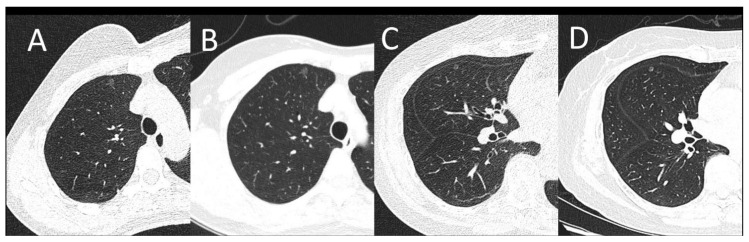
(**A**,**B**): A female patient with a BMI of 20.31 kg/m^2^. A subpleural pulmonary nodule was shown on LDCT (**A**) and SDCT at the same location as A (**B**). (**C**,**D**): A female patient with a BMI of 24.91 kg/m^2^. One pulmonary nodule can be seen on LDCT (**C**) and SDCT at the same location as C (**D**). Images of both protocols can clearly show the lesion. The image noise was appreciably increased on LDCT images, however, there was sufficient contrast between the nodule and the adjacent lung tissue for lesion identification.

**Figure 3 diagnostics-13-03235-f003:**
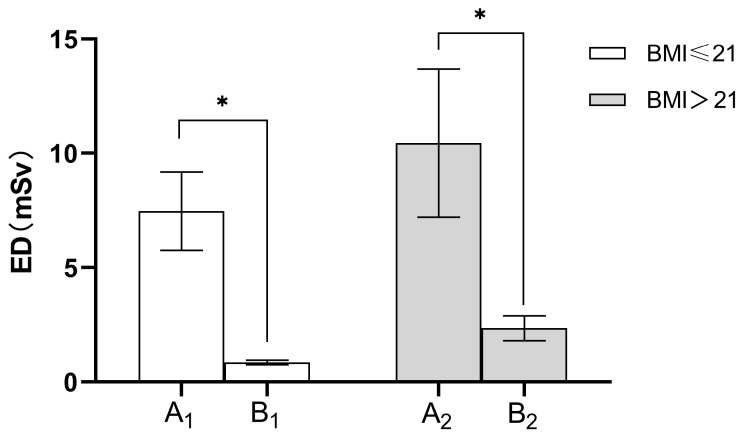
Comparison of average ED values in various groups. A1 = SDCT group (BMI ≤ 21 kg/m^2^); B1 = LDCT group (BMI ≤ 21 kg/m^2^); A2 = SDCT group (BMI > 21 kg/m^2^); B2 = LDCT group (BMI > 21 kg/m^2^). The effective dose (ED) decreased by an average of 89.42% (for BMI ≤ 21 kg/m^2^), 77.68% (for BMI > 21 kg/m^2^), respectively, under the LDCT protocol compared to the SDCT protocol. * indicates *p* < 0.05, Bonferroni corrected.

**Table 1 diagnostics-13-03235-t001:** Demographics and baseline nodule characteristics.

Basic Information	Group A1 (*n* = 32)	Group B1 (*n* = 34)	*p* Value	Group A2 (*n* = 51)	Group B2 (*n* = 67)	*p* Value
Gender (F/M)	17/8	22/6	0.384	25/14	37/21	0.975
Age (Years)	56.96 ± 11.05	52.93 ± 11.93	0.198	56.54 ± 8.99	57.89 ± 9.00	0.463
BMI (kg/m^2^)	20.58 ± 0.99	20.11 ± 1.33	0.154	24.50 ± 1.72	25.37 ± 2.55	0.064
Lesion Type						
pGGN	20 (62.5%)	19 (55.9%)		34 (66.7%)	34 (50.7%)	
psGGN	11 (34.4%)	14 (41.2%)		13 (25.5)	29 (43.3%)	
Solid	1 (3.1%)	1 (2.9%)		4 (7.8%)	4 (6.0%)	
Lesion Size (mm)						
Lesion ≤ 5	4 (12.5%)	4 (11.8%)		10 (19.6%)	10/(14.9%)	
5 < Lesion ≤ 10	20 (62.5%)	20 (58.8%)		29 (56.9%)	41 (61.2%)	
10 < Lesion ≤ 20	8 (25.0%)	10 (29.4%)		12 (23.5%)	16 (23.9%)	
Lesion Location						
RUML	8 (25.0%)	9 (26.5%)		13 (25.5%)	24 (35.8%)	
RLL	7 (21.9%)	9 (26.5%)		11 (21.6%)	13 (19.4%)	
LUL	9 (28.1%)	3 (8.8%)		16 (31.3)	18 (26.9%)	
LLL	8 (25.0%)	13 (38.2%)		11 (21.6%)	12 (17.9%)	

BMI, body mass index; pGGN, pure ground glass nodule; psGGN, part-solid GGN; RUML, right upper MIDDLE lobe; RLL, right lower lobe; LUL, left upper lobe; LLL, left lower lobe.

**Table 2 diagnostics-13-03235-t002:** Procedure characteristics of localization.

Group	Nodule Density (HU)	Nodule Size (mm)	Patient Position Supine/Prone/Lateral (n)	Nodule Distance to Pleural Surface (mm)	Nodule Distance to Needle Tip (mm)	Needle Insertion Depth from Pleura (mm)	Pneumothorax (n)	Bleeding (n)	Number of Needle Insertions (n)	Localization Procedure Duration (min)	Surgery Duration (h)
Group A1	−548.93 ± 202.82	8.82 ± 3.39	3/10/19	16.21 ± 12.72	15.01 ± 13.45	26.63 ± 13.85	9 (25)	6 (25)	1.76 ± 1.27	9.56 ± 4.37	1.38 ± 0.44
Group B1	−569.60 ± 139.36	8.59 ± 2.98	4/7/23	19.09 ± 13.54	15.13 ± 11.69	32.15 ± 16.65	6 (28)	7 (28)	1.62 ± 0.78	9.43 ± 2.57	1.21 ± 0.40
*p* value	0.629	0.773		0.377	0.969	0.108	0.360	1.000	0.635	0.894	0.154
Group A2	−518.66 ± 222.88	8.29 ± 3.78	9/15/27	21.05 ± 14.95	19.97 ± 18.16	30.24 ± 12.49	15 (39)	13 (39)	2.15 ± 1.44	10.49 ± 4.08	1.55 ± 0.63
Group B2	−460.22 ± 223.05	8.31 ± 2.83	10/21/36	22.83 ± 16.61	16.69 ± 12.10	29.15 ± 14.48	13 (58)	19 (58)	2.05 ± 1.39	10.47 ± 4.58	1.43 ± 0.42
*p* value	0.161	0.963		0.550	0.243	0.668	0.111	1.000	0.735	0.988	0.298

Group A1 = SDCT group (BMI ≤ 21 kg/m^2^); Group B1 = LDCT group (BMI ≤ 21 kg/m^2^); Group A2 = SDCT group (BMI > 21 kg/m^2^); Group B2 = LDCT group (BMI > 21 kg/m^2^).

**Table 3 diagnostics-13-03235-t003:** Lesion conspicuities for lung nodules.

Group	Nodule Density	Nodule Size
Group B1	−546.64 ± 136.67	8.59 ± 2.98
Group C1	−544.91 ± 134.27	8.75 ± 3.10
*p* value	0.458	0.135
Group B2	−460.22 ± 223.05	8.31 ± 2.83
Group C2	−443.68 ± 225.56	8.45 ± 2.86
*p* value	0.108	0.350

Group B1 = LDCT group (BMI ≤ 21 kg/m^2^); Group C1 = SDCT group (BMI ≤ 21 kg/m^2^); Group B2 = LDCT group (BMI > 21 kg/m^2^); Group C2 = SDCT group (BMI > 21 kg/m^2^).

**Table 4 diagnostics-13-03235-t004:** Subjective overall image qualities.

Group	Score	
	BMI ≤ 21 kg/m^2^	BMI > 21 kg/m^2^
Group A	4.92 ± 0.277	4.92 ± 0.270
Group B	3.86 ± 0.356	3.95 ± 0.223
*p* value	0.000	0.000

Group A = SDCT; Group B = LDCT.

**Table 5 diagnostics-13-03235-t005:** Radiation Dose in Each Group.

Group	CTDI (mGy)	DLP (mGy.cm)	ED (mSv)
Group A1	5.24 ± 0.95	533.58 ± 122.06	7.47 ± 1.71
Group B1	0.56 ± 0.00	56.86 ± 4.73	0.79 ± 0.07
*p* value	<0.001	<0.001	<0.001
Group A2	6.69 ± 1.47	746.01 ± 230.91	10.44 ± 3.23
Group B2	1.48 ± 0.00	167.02 ± 38.76	2.33 ± 0.54
*p* value	<0.001	<0.001	<0.001

Group A1 = SDCT group (BMI ≤ 21 kg/m^2^); Group B1 = LDCT group (BMI ≤ 21 kg/m^2^); Group A2 = SDCT group (BMI > 21 kg/m^2^); Group B2 = LDCT group (BMI > 21 kg/m^2^).

## Data Availability

Not applicable.

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
