# Peer review of "Implementation of Individualized Low-Dose Computed Tomography-Guided Hook Wire Localization of Pulmonary Nodules: Feasibility and Safety in the Clinical Setting"

_diagnostics, 2023, doi:10.3390/diagnostics13203235_

Round 1

Reviewer 1 Report

need to add more surgical data, like hook-wire fall-off rate, surgery duration

Reviewer 2 Report

This is a very well structures article. Great work. To my understanding, the originality of the article is the calculation of dose and the comparison between the standard and the low dose application. Vary useful and interesting results.

Minor comments:

L116: "... CT images were obtained in order to determine the shape, size, and location...", change to: "... CT images were obtained to determine the shape, size, and location..."

Reviewer 3 Report

Well structured and thorough study. Detailed methodology and statistical analysis.  Limitations are clearly stated. 

Round 2

Reviewer 1 Report

the fall-out rate or setting success rate is quiet important for an article named feasibility and safety 

I am not an English native